# Biological Evaluation of 3-Benzylidenechromanones and Their Spiropyrazolines-Based Analogues

**DOI:** 10.3390/molecules25071613

**Published:** 2020-04-01

**Authors:** Angelika A. Adamus-Grabicka, Magdalena Markowicz-Piasecka, Marcin Cieślak, Karolina Królewska-Golińska, Paweł Hikisz, Joachim Kusz, Magdalena Małecka, Elzbieta Budzisz

**Affiliations:** 1Department of Cosmetic Raw Materials Chemistry, Faculty of Pharmacy, Medical University of Lodz, Muszynskiego 1, 90-151 Lodz, Poland; angelika.adamus@umed.lodz.pl; 2Laboratory of Bioanalysis, Department of Pharmaceutical Chemistry, Drug Analysis and Radiopharmacy, Medical University of Lodz, Muszynskiego 1, 90-151 Lodz, Poland; magdalena.markowicz@umed.lodz.pl; 3Centre of Molecular and Macromolecular Studies, Polish Academy of Sciences, Sienkiewicza 112, 90-363 Lodz, Poland; marcin@cbmm.lodz.pl (M.C.); kkrolews@cbmm.lodz.pl (K.K.-G.); 4Department of Molecular Biophysics, Faculty of Biology and Environmental Protection, University of Lodz, Pomorska 141/143, 90-236 Lodz, Poland; pawel.hikisz@biol.uni.lodz.pl; 5Institute of Physics, University of Silesia, Uniwersytecka 4, 40-007 Katowice, Poland; joachim.kusz@us.edu.pl; 6Department of Physical Chemistry, Theoretical and Structural Chemistry Group, Faculty of Chemistry, University of Lodz, Pomorska 163/165, 90-236 Lodz, Poland; magdalena.malecka@chemia.uni.lodz.pl

**Keywords:** 3-benzylideneflavanone/3-benzylidenechromanone, spiropyrazolines, cytotoxicity, lipophilicity, DNA interaction

## Abstract

A series of 3-benzylidenechrmanones **1**, **3**, **5**, **7**, **9** and their spiropyrazoline analogues **2**, **4**, **6**, **8**, **10** were synthesized. X-ray analysis confirms that compounds **2** and **8** crystallize in a monoclinic system in P2_1_/n space groups with one and three molecules in each asymmetric unit. The crystal lattice of the analyzed compounds is enhanced by hydrogen bonds. The primary aim of the study was to evaluate the anti-proliferative potential of 3-benzylidenechromanones and their spiropyrazoline analogues towards four cancer cell lines. Our results indicate that parent compounds **1** and **9** with a phenyl ring at C2 have lower cytotoxic activity against cancer cell lines than their spiropyrazolines analogues. Analysis of IC_50_ values showed that the compounds **3** and **7** exhibited higher cytotoxic activity against cancer cells, being more active than the reference compound (4-chromanone or quercetin). The results of this study indicate that the incorporation of a pyrazoline ring into the 3-arylideneflavanone results in an improvement of the compounds’ activity and therefore it may be of use in the search of new anticancer agents. Further analysis allowed us to demonstrate the compounds to have a strong inhibitory effect on the cell cycle. For instance, compounds **2**, **10** induced 60% of HL-60 cells to be arrested in G2/M phase. Using a DNA-cleavage protection assay we also demonstrated that tested compounds interact with DNA. All compounds at the concentrations corresponding to cytotoxic properties are not toxic towards red blood cells, and do not contribute to hemolysis of RBCs.

## 1. Introduction

The chromanones and their flavanone derivatives comprise an interesting group of compounds because they easily react with nucleophilic reagents, the most investigated of these reactions being those which take place with hydrazine or its analogues, resulting in the production of a five-membered heterocyclic ring. Pyrazolines are heterocyclic compounds having two adjacent atoms within the ring. They have only one endocyclic double bond and are basic in nature. The dihydro derivative of pyrazole is known as pyrazoline. Depending on the position of the double bond it can exist in three separate forms: Δ^1^-pyrazoline (the double bond exists between the first and second nitrogen atoms), Δ^2^-pyrazoline (the double bond occurs between the second nitrogen atom and the third carbon atom) and Δ^3^-pyrazoline (the double bond occurs between the third and fourth carbon atom) [1]. As their chemical structure is based on the presence of electron-rich nitrogen heterocycles, they display important biological activities [2]. Pyrazoline derivatives widely occur in nature in the form of alkaloids, pigments and vitamins. Due to the anticancer (Figure 1A), antibacterial (Figure 1B), antifungal (Figure 1C), anti-inflammatory (Figure 1D), and antidepressant (Figure 1E) properties of spiropyrazole analogs, many studies have examined their biological activities [3,4,5]. Certain derivatives, such as nitric oxide synthase (NOS) inhibitor (Figure 1F) and cannabinoid CB1 receptor antagonist (Figure 1G), also demonstrate potent selective biological activity against certain receptors [6,7]. Due to their biological activity, they have been used in the treatment of various diseases.

Over the last thirty years, many methods have been developed for the synthesis of pyrazolines and their analogues; however, while numerous such analogues have been described in the literature, the 4,5-dihydro-1*H*-pyrazolines are the most often studied. Their best known method of synthesis is based on the reaction of α,β-unsaturated ketones with hydrazines, which results in the formation of a hydrazone, which is cyclized to form a 2-pyrazoline [8]. Much attention has been focused on the synthesis of 3-arylidenechromanones/flavanones, which are precursors in the reaction for obtaining 3-spiropyrazolines; these were synthesized by reacting 3-benzylidenechromanones or 3-benzylidene-flavanones with diazomethane in anhydrous acetone [9]. According to the Toth [9] the classical synthesis of 3-arylideneflavanones/chromanones with diazomethane allows only one stereoisomer of spiro-1-pyrazolines to be obtained.

Following on from a previous paper [10], the present study broadens our studies of the reactivity of 3-arylideneflavanones and 3-arylidenechromanone with diazomethane on a larger number of compounds containing various benzylidene substituents. The formation of spiro- compounds as a result of attack by a nucleophilic reagent on the carbon of the 3-arylideneflavanone most probably depends on the degree of electropositivity of the carbon; this would govern reactions with the various resonance structures of diazomethane [9]. It is possible that the resonance structure of diazomethane, with a partial negative charge on the carbon atom, may be attacked in the reaction of the formation of 3-spiropyrazolines (Scheme 1).

The aim of this study is to compare the cytotoxic activities of a series of chromanones/flavanones and spiropyrazoline analogues as substrates and products of their modification reactions with diazomethane. The compounds were tested for biological activity on four cell lines: HL-60 (human leukemia cell line), NALM-6 (human peripheral blood leukemia cell line), WM-115 (melanoma cell line) and COLO-205 (human colon adenocarcinoma cells). It also examines the influence of various substituents on the lipophilicity and cytotoxic effect of the compounds, as well as their interaction with DNA and cell cycle arrested studies. Moreover, we searched for the less toxic compounds compared to references compounds. Both the 3-benzylidene and spiro- analogues were digested of plasmid DNA with BamHI restriction nuclease and the study of cell cycle phases using a flow cytometer has been done.

Correct design and systematic examination of the structure-activity relationship of pyrazoline compounds appears to be a good strategy for new drug design. The obtained compounds may be a valuable basis for discovering safe potent drug candidates with lesser side effects.

## 2. Results and Discussion

Two structures, benzylideneflavanone and benzylidenechromanone substituted with diethylamine group, have been examined previously in a study that compared the exacytotoxic activity of the two compounds and checked whether they destroy erythrocyte membranes [10]. Encouraged by these positive results, the present manuscript examines the relationship between the biological activity of the tested compounds and their chemical structure: for this purpose ten 3-arylidenechromanone analogues with a Δ^1^-pyrazoline ring were synthesized and their physicochemical characteristics, together with their cytotoxicity, determined. All synthesized compounds were tested for their compatibility with blood and whether they influenced the integrity of the erythrocyte membrane. Analogues **2**, **7**, **10**, **11**, and **12** exhibited the greatest cytotoxicity towards the tested cancer cell lines and thus were taken for further study. The next step examined whether the compounds would inhibit digestion of plasmid DNA in a similar way to a reference compound (daunorubicin); in addition, the compounds were subjected to DNA interaction studies to identify the phase of the cell cycle they would block. Finally, two analogues, **2** and **8**, were chosen for further testing to describe their crystal structure and characterize their intermolecular interactions in the crystal lattice.

### 2.1. Chemistry

The synthesis of the described compounds has been divided into two stages. In the first step the compounds **1**, **3**, **5**, **7** and **9** were prepared by a conventional reaction between 2-phenylchroman-4-one or chroman-4-one and the appropriate aryl aldehyde. The exocyclic α,β-unsaturated ketones were synthesized in the piperidine medium; in a second step, these unsaturated ketones were reacted with an ethereal solution of diazomethane in anhydrous acetone to produce spiro-1-pyrazolines **2**, **4**, **6**, **8**, **10** in a regioselective and stereospecific manner (Scheme 1). The structures of all compounds were characterized using IR, ^1^H- and ^13^C-NMR, MS spectroscopy and elemental analysis. A detailed description of the spectra of the tested compounds is provided in the Appendix A with the ^1^H- and ^13^C-NMR spectra (see Appendix A). Part 3 (Materials and Methods) presents a detailed description of the IR bands and signal shifts of the spectra.

### 2.2. X-ray Single-Crystal Structure of ***2*** and ***8***

The molecular structure of compounds **2** and **8** is shown in Figure 2. Molecules of **2** and **8** crystalize in a monoclinic system in P2_1_/n; however, in **8**, an asymmetric unit has three molecules. The main chroman skeleton in **2** consists of a benzene and a pyran ring with a phenyl substituent at the C2 atom. In addition, 4-(4-methoxyphenyl)-4,5-dihydro-3*H*-pyrazole is attached at the C3 position. 

Molecule **8** contains a chroman skeleton with a 4-phenyl-4,5-dihydro-3*H*-pyrazole at position C3. The pyran ring and 4,5-dihydro-3*H*-pyrazole adopt an envelope conformation in both structures, with puckering parameters and asymmetry parameters given in Table 1 [11,12].

The crystal packing of **2** and **8** is enhanced by a network of hydrogen bonds (Figure 2). The molecules of compound **2** are linked by four intermolecular hydrogen bonds: C12-H12A…O1 (1/2−x, 1/2 + y, 1/2 − z), C2-H2…O4 (1/2 − x,−1/2 + y,1/2 − z), C26-H26…N2 (1/2 − x, 1/2 + y, 1/2 − z), C11-H11…N1 (1/2 − x, 1/2 + y, 1/2 − z), which are responsible for the formation of dimers (Figure 3). 

Additionally, each molecule is linked by a C24-H24…O4 (1 + x, y, z) hydrogen bond forming chains propagating along *a* axis. The intermolecular hydrogen bonds in crystal structure **8** produce a 3-D net of hydrogen bonds (Figure 4). The geometric parameters for the hydrogen bonds are presented in Table 2. Additionally, it is worth mentioning that compound **2** has higher values for all tested cytotoxic activities compared to **8** (Table 3); this is associated with its more extended spatial structure, in which a bulky phenyl substituent is attached in the C2 position. A similar trend is observed for benzylideneflavanone and benzylidenechromanone analogues [13]. However, taking into account the structure **2** and **10**, which are similar and differ only the position of methoxy group in phenyl fragment cytotoxic activity is different. Therefore it could be concluded that in case of COLO-205 the position of OCH_3_ group influences on the activity (higher when m-OCH_3_ group is attached). This trend is also visible if we compare **2**, **4** and **10** structures, for *p*-OCH_3_ substituent cytotoxic activity (COLO-205) is lower that for m-OCH_3_ substituent.

### 2.3. Experimental Lipophilicity of the Synthesized Flavone Analogues

Lipophilicity is one of the most important physicochemical properties in pharmaceutical research. It can be considered a key determinant of the pharmacokinetic properties of a drug. It is commonly measured by the partition coefficient between water and *n*-octanol, being expressed as log*P* [14]. The coefficient log*P* shows the affinity of a molecule for a lipophilic environment [15,16]. Knowledge of the partition coefficient is valuable: it is frequently used in structure–activity relationship (SAR) and quantitative structure–activity relationship (QSAR) studies [17,18]. The experimental lipophilicity of the synthesized compounds, *viz*. benzylideneflavanones, benzylidenechromanones and spiropyrazolines, was determined using RP-TLC. The log*P* results indicate that the tested compounds have lipophilic character. The log*P* values ranged from 4.08 to 5.88 for all tested compounds except **8** (log*P* = 2.35), indicating a possible relationship between lipophilicity and chemical structure. Compound **2** demonstrated some dependence in terms of spatial arrangement, and greater cytotoxicity than **8** towards four cell lines (Table 3). This can be attributed to its more spatial structure, where a bulky phenyl substituent is attached in the C2 position. A similar trend can be seen for benzylideneflavanone analogues: all compounds with a phenyl ring in the C2 position have higher log*P* values. It was also found that lipophilicity of a compound depends on its chemical structure: the presence of a phenyl or methoxy substituent increases the log*P* value, and unsubstituted compounds have lower lipophilicity (**8**). Compound **12** has a lower log*P* value (3.43) than **11** (5.69) (Figure 5). It is important to examine the relationship between structure of the compound and its lipophilicity and cytotoxicity. Compound **11**, with a phenyl ring at C2, exhibits greater biological activity than the unsubstituted benzylidenechromanone. The biological activity of the compounds increases with lipophilicity, but it cannot exceed the maximum value. According to Lipinski’s rule and drug likeliness parameters, compounds with higher lipophilicity, up to a maximum log*P* value of 5, may be good candidates for drugs [19,20]. The compounds with log*P* greater than 3 present high lipophilicity and high potential for bioaccumulation in cells.

### 2.4. Cytotoxicity toward Human Cancer Cells

Modern medicine is still struggling with the problem of fully effective chemotherapy. Therefore, there is a strong need to develop and synthesize new chemotherapeutics characterized by adequate effectiveness, selectivity and specificity of action against cancer cells. The need to identify new compounds with effective anticancer activity, and high selectivity against cancer cells and low toxicity to normal cells, demands the testing of a wide variety of candidates. Undoubtedly, the turning point for the development of research on organometallic complexes was the discovery of cisplatin in the 20th century, which is used to treat lung, ovarian, neck or head cancer, among others. However, its effective biological anticancer activity is unfortunately associated with high systemic cytotoxicity, particularly myelosuppression and nephro- and hepatotoxicity, which is a serious factor limiting its wide use.

Our present study evaluated the anticancer activity of a large group of 3-arylidenechromanone, 3-arylideneflavanone, spirochromanone and spiroflavanone analogues against four tumor lines. The experiments serve as targeted basic research and are aimed at gaining new knowledge related to the anticancer activity of selected 3-benzylidenechromanone/3-benzylideneflavanone analogues and their pyrazolines.

The first stage of the study used the MTT viability test to determine the cytotoxicity of ten compounds, i.e., 3-arylidenechromanone, 3-arylideneflavanone, spirochromanone and spiro- flavanone analogues, against four human cancer cell lines: HL-60, NALM-6, WM-115 and COLO-205. In addition, and 4-chromanone, quercetin, cisplatin and carboplatin were included in the tests as reference compounds. The IC_50_ values for the entire series of tested compounds are given in Table 3.

Our studies indicate that the antiproliferative activity of the test compounds was determined by their chemical structure. Compounds **2**, **7** and **10** demonstrated the greatest antiproliferative activity against cancer cells. As expected, they demonstrated dose-dependent inhibition of human cancer cell line growth with a significant decrease in cell viability being associated with increasing concentrations. The most active compound was **2**, with IC_50_ < 10 µM against three cancer cell lines: HL-60, NALM-6 and WM-115; however, COLO 205 proliferation was slightly less inhibited (IC_50_ 29.3 µmol/L). 

Compound **7** demonstrated promising antiproliferative properties, especially towards NALM-6 and WM-115 cell lines, with IC_50_ is equal to 6.83 and 12.75 µmol/L respectively. It is worth mentioning that **10** also demonstrated noticeable cytotoxic effects against HL-60 and NALM-6 cells (IC_50_ about 9.4 and 25 µmol/L respectively); however, WM-115 and COLO-205 were markedly more resistant. On the basis of these results, further studies were performed on the molecular mechanisms of the anticancer activity of the complexes against the corresponding cell lines.

It is also worth mentioning that in further studies we also used two additional compounds: (*E*)-3-(4-*N*,*N*-diethylaminobenzylidene)chroman-4-one (**12**) and (*E*)-3-(4-*N*,*N*-diethylaminobenzylidene)-2-phenylchroman-4-one (**11**). Their synthesis, structure and promising antiproliferative properties against cancer cells (HL-60, NALM-6 and WM-115) have been described previously [10]. Both compounds showed high cytotoxic activity against the HL-60 cell line: 11.76 µM **12** and 8.36 µM **11**. The IC_50_ values for NALM-6 cell line were similar: 8.69 µM **12** and 9.08 µM **11**. Only for WM-115 cell line the cytotoxicity values were not similar. The compound **11** exhibited high cytotoxic activity with an IC_50_ value of 6.45 µM, i.e., three times lower than for **12**. Hence, both compounds appear to exhibit selectivity for human leukemia cell lines.

### 2.5. Red Blood Cell Lysis Assay

All synthesized compounds with potential biological activity have to be tested for blood compatibility since xenobiotics will react with blood cells upon administration. In this case, biocompatibility refers to the quantification of cellular and plasma components of the blood. The measurement of the integrity of RBC membrane is considered a simple and reliable method for estimating hemocompatibility [21,22]. The effects of the synthesized compounds, together with those of the two reference compounds flavanone and chromanone, on RBC hemolysis are depicted in Figure 6A,B. Neither flavanone nor chromanone was found to have any unfavourable effect on the erythrocyte membrane over the whole concentration range. A statistically significant increase in the rate of haemolysis was observed for most of the tested compounds at the higher concentrations. For instance, compounds **3** and **4** contributed to a significant disintegration of the erythrocyte membrane at concentrations above 25 µM, manifested by a higher percentage of hemolysis in comparison to control. However, these values did not exceed 5%, which means that there is not a risk of pathological hemolysis in in vivo conditions. Similar results were obtained for compounds **5**, **6**, **9**, and **10**.

In the case of compound **7**, no significant effects on RBC hemolysis were observed up to 35 µM; however, a significantly greater disintegration of RBC membrane was reported at concentrations of 250 µM and 500 µM in the related compound **8**. The differences in the tested concentrations of these compounds stem from the fact that they affect the cellular growth (Table 3) at different concentration ranges.

### 2.6. Red Blood Cell Morphology

Studies on erythrocyte morphology complement the erythrotoxicity studies because they provide information on potential immediate interactions of the tested compounds with the erythrocyte lipid bilayer, and the possibility of creating pathological forms of erythrocytes, which are not able to carry oxygen. The effects of the synthesized compounds, and the two reference compounds chromanone and flavanone, on the morphology of erythrocytes is shown in Figure 6. Briefly, 2% RBC suspension was incubated with 0.9% NaCl (control samples) or tested compounds at various concentrations corresponding to their activity towards cancer cell lines. Dyscocytes were observed in control samples, while echinocyte formation was observed in cultures treated with chromanone or flavanone over the entire concentration range (Figure 6A). 

Echinocytes are formed when the agents that preferentially accumulate in the outer layer of the RBC membrane bilayer expand it, thus giving the cell an echinocytic shape [21]. In most samples, the population of echinocytes exceeded 80% of the total erythrocytes observed in the microscopic field of view. Incubation of erythrocytes with compound **1** led to the extensive formation of echinocytes and anisocytes, while compound **2** resulted mainly in echinocytosis. In addition, echinocytes accounted for the major part of erythrocytes following treatment with compounds **3** and **4** at a concentration of 100 µmol/L, and for compounds **5** and **6**, as well as **9** and **10**. Compound **7** primarily led to echinocytosis, however single anisocytes could also be detected. In summary, in vitro exposure of RBCs to different concentrations of examined compounds and reference molecules resulted mostly in the formation of echinocytes. Nevertheless, the process of transformation of biconcave erythrocyte to echinocyte occurs naturally in blood vessels, and results from the insertion of xenobiotics into the outer monolayer of the erythrocyte membrane. As this is a reversible process [22] caused by a series of chemical and physical factors including increased ion strength, alkaline pH and decreased ATP level, it may be assumed that most of the analyzed compounds do not affect RBCs in a pathological manner at the relevant concentrations.

### 2.7. Digestion of Plasmid DNA with BamHI Restriction Nuclease

Some of the test compounds showed significant toxicity toward cancer cells. For example compounds **2** or **7** demonstrated IC_50_ values in the single micromolar range (Table 3). This cytotoxicity was tested with regard to its effect on the cell cycle and the ability to modify DNA. In addition to the compounds shown in Table 4, two highly-cytotoxic compounds (**11** and **12**) were also tested. Their structure and biological activity were described in a previous article [10]. The chemical structures of the test compounds suggests that they may intercalate with double-stranded nucleic acids. To test this hypothesis, plasmid DNA was incubated with the test compounds and subsequently digested with BamH1 endonuclease. As expected, daunorubicin (positive control) which is a strong DNA intercalator, prevented linearization of DNA (Figure 7, lane B). Interestingly, chromanone/flavanone test compounds also inhibited digestion of plasmid DNA (Figure 7, lanes C -I). This conclusion is supported by the presence of the circular form of plasmid DNA after digestion with BamH1. These results suggest that chromanone/flavanone analogues intercalate to genomic DNA.

### 2.8. Cell Cycle Analysis

The next stage examined the effects of the selected compounds on cell cycle progression in HL-60 leukemia cells. The five chromanone/flavanone compounds and quercetin (reference) were tested at concentrations of IC_50_ and 0.5 × IC_50_. The cell cycle distribution of HL 60 cells after treatment with test compounds is shown in Figure 8 and Table 4. As expected, in the presence of quercetin, about 50–60% of cells were blocked in the G2/M phase (see Appendix A). Interestingly, the test compounds also induced mitosis block in G2/M. Only **7** moderately inhibited cell cycle progression, arresting 20–26% of cells. The effect of **2**, **10**, **11** and **12** was very pronounced and similar to quercetin. In the presence of these compounds, about 50–60% of cells were arrested in G2/M. These results confirm that test compounds have a strong influence on the cell cycle and arrest the HL-60 cells in the G2/M phase.

## 3. Materials and Methods

### 3.1. General Information

The compound pairs **1** and **2**, **5** and **6** and **9** and **10** were synthesized according to Pijewska [8]. Compounds **3** and **7** were synthesized according to Levai and Schag [23]. Compounds **4** and **8** were synthesized according to Toth and Szollosy [9].

All compounds were purified by crystallization from methanol. All solvents (methanol, ethanol, toluene) used in this work were purchased from Sigma-Aldrich (St. Louis, MO, USA) and Polish Chemical Reagents (Gliwice, Poland), and used without further purification. Melting points were determined on a B-540 Melting Point apparatus (Büchi, Flawil, Switzerland) in capillary mode and they were uncorrected. The infrared transmission spectra of the crystalline products were recorded at the University of Lodz Faculty of Chemistry using a Nexus FT-IR spectrophotometer (Thermo Nicolet, Waltham, MA, USA). The MS-ESI were measured at the University of Lodz Faculty of Chemistry on a 500–MS LC Ion Trap mass spectrometer (Varian, Palo Alto, CA, USA). Accurate mass measurements of product ions were confirmed using a Waters (Manchester, UK) q-TOF Synapt quadrupole time-of-flight hybrid mass spectrometer. Elemental analyses were performed at the Faculty of Chemistry (University of Lodz) using a Vario Micro Cube analyzer by Elemental (Langenselbold, Germany). ^1^H- (600 MHz) and ^13^C-NMR spectra (150 MHz) were recorded at the University of Lodz Faculty of Chemistry on an Avance III 600 MHz instrument (Bruker, Billerica, MA, USA). The samples were dissolved in deuterated DMSO and CDCl_3_. Chemical shifts are given in ppm, coupling constants in Hz. Chemical shifts are referenced to the residual solvent signals, 2.50 ppm for ^1^H in DMSO-*d*_6_, 7.26 ppm for ^1^H in CDCl_3_ and 39.5 ppm for ^13^C in DMSO-*d*_6_, 77.0 ppm for ^13^C in CDCl_3_. Computational studies were used to predict the properties and future use of potential drugs. Molinspiration Cheminformatics was used for the calculation of important molecular properties, molecular processing, bioactivity and the high-quality depiction of new molecules. Lipophilicity was determined by RP-TLC with small amounts of organic solvents. The erythrotoxicity of the compounds was evaluated in Laboratory of Bioanalysis in the Department of Pharmaceutical Chemistry, Drug Analysis and Radiopharmacy (Medical University of Lodz). The erythrocyte morphology was evaluated using an Opta-Tech phase-contrast microscope (Opta-Tech, Warsaw, Poland) with OptaView 7 software.

Diazomethane was prepared for the synthesis spiropyrazolines analogues as follows: Briefly, 50% aqueous potassium hydroxide solution (30 mL) and 100 mL diethyl ether was placed in a 500 mL flat-bottomed flask. The solution was cooled in an ice-salt mixture to 5 °C and 0.2 M of *N*-nitrosomethylurea (10.3 g) was added in portions. The mixture was stirred. The solution turned yellow and was placed in a previously-cooled separatory funnel. The solution was separated and the ether layer of the diazomethane was placed in a flask covered with pellets of potassium hydroxide. The aqueous layer was placed into a flask with an aqueous solution of benzoic acid.

### 3.2. Synthesis of (E)-3-(4-methoxybenzylidene)-2-phenylchroman-4-one (***1***)

A mixture of 0.01 mol (2.243 g) 2-phenylchroman-4-one, 0.01 mol (1.362 g) 4-methoxy-benzaldehyde and five drops of piperidine were heated in oil bath with mechanical stirring at 130 °C for five hours. The progress of the reaction was tested by TLC (toluene:methanol 9:1). The mixture was left to cool at room temperature and dissolved in methanol. After 24 h, the compound was precipitated and then purified by crystallization from methanol. The compound **1** was obtained as a white powder. Yield: 41% m.p: 139.8–141.3 °C. MS (ESI+): m/z 343.3 C_23_H_18_O_3_ [M + H]^+^. IR (KBr) ν(cm^−1^): 3030 (C-H aromat), 2945, 2910 (C-H aliph.), 1675 (C=O), 1602, 1578, 1506 (C=C), 1304 (C-O), 1172 (C-O), 1148 (C-O-C). ^1^H-NMR (DMSO-d_6_) δ (ppm): 2.28 (3H, s, OCH_3_), 6.74 (1H, s, C2-H), 7.99 (1H, s, =CH), 7.04–7.81 (13H, m, C-H arom.) (Appendix A). ^13^C-NMR (CDCl_3_) δ (ppm): 29.7 (OCH_3_), 77.9 (C2-H), 116.2, 118.6, 121.7, 126.0, 127.6, 127.7, 128.6, 132.5, 136.0, 138.1, 140.0 (CH_arom_, =CH), 122.3, 132.5, 158.6, 158.9 (C_arom_), 183.0 (C=O) (Appendix A). Anal. Calc. For C_23_H_18_O_3_ (M = 342.38 g/mol) % C: 80.68; %H: 5.26. Found %C: 80.60; % H: 5.25.

### 3.3. Synthesis of 5′-(4-methoxyphenyl)-2-phenylo -4′,5′-dihydro-4H-spiro [chromano-3,3′-pirazol]-4-one (***2***)

Briefly, 2.5 mmol (0.856 g) of compound **1** was dissolved in anhydrous acetone (5 mL) to give a light yellow solution. The flask containing the solution was placed in the ice bath and an ethereal solution of diazomethane (10 mmol) was added in excess. The mixture in the flask was left in the freezer for 48 h. The yellow precipitate was filtered off and purified by column chromatography and crystallized from methanol. Yield: 34% m.p. 128–129 °C. IR (KBr) ν(cm^−1^): 3029 (C-H aromat.), 2958, 2837 (C-H aliph.), 1677 (C=O), 1605, 1580, 1547, 1512, 1472 (C=C, N=N), 1307 (C-N), 1240 (C-O), 1146 (C-O-C). ^1^H-NMR (DMSO-d_6_) δ (ppm): 3.74 (3H, s, OCH_3_), 4.01 (1H, dd, *J*_AB_ = 7.6 Hz, *J*_BX_ = 6.9 Hz, CH), 4.14 (1H, dd, *J*_AB_ = 12.27 Hz, *J*_BX_ = 7.15 Hz CH_2_), 4.23 (1H, dd, *J*_AB_ = 12.27 Hz, *J*_BX_ = 1.96 Hz CH_2_), 5.86 (1H, s, C2-H), 6.87–7.68 (13H, m, C-H aromat) (Appendix A). ^13^C-NMR (CDCl_3_) δ (ppm): 23.6 (OCH_3_), 55.2 (CH), 77.3 (CH), 79.5 (C2-H), 112.4, 113.9, 118.2, 119.2, 123.4, 127.3, 127.4, 128.5, 129.8, 138.9 (CH_arom_), 121.3, 135.7, 142.6, 152.8, 158.7, 159.6 (C_arom_), 186.3 (C=O) (Appendix A). Anal. Calc. For C_24_H_20_O_3_N_2_ (M = 384.43 g/mol) %C: 74.98; %H: 5.20; %N: 7.34. Found %C: 74.07; %H: 5.21; %N: 7.43. MS (ESI+): m/z 385.4 C_24_H_20_O_3_N_2_ [M + H]+. HRMS (ESI+): *m*/*z* [M + H]^+^ calcd. for C_24_H_21_O_3_N_2_: 385.1552; found: 385.1548 (see Appendix A).

### 3.4. Synthesis of (E)-3-(4-methoxybenzylidene)chroman-4-one (***3***)

A mechanically-stirred mixture of 0.01 mol (1.481 g) of chroman-4-one, 0.01 mol (1.362 g) of *p*-methoxybenzaldehyde and five drops of piperidine were heated at 150 °C in an oil bath for four hours. After cooling, the reaction mixture was left for 24 h at room temperature. The solidified product was filtered and crystallized from methanol. Compound **3** was obtained as a cream-colored powder. Yield: 55% m.p: 133.8–134.85 °C. MS (ESI+): m/z 267.5 C_17_H_14_O_3_ [M + H]+. IR (KBr) ν(cm^−1^): 3038, 3000 (C-H aromat.), 2958, 2866 (C-H aliph.), 1665 (C=O), 1603, 1568, 1510, 1477,1463 (C=C), 1146 (C-O-C), 1112 (C-O). ^1^H-NMR (CDCl_3_) δ (ppm): 1.58 (3H, s, OCH_3_), 3.87 (1H, s, =CH), 5.38 (2H, d, *J*_AB_ = 16.60 Hz C2-H), 6.96–8.03 (8H, m, C-H aromat.) (see Appendix A). ^13^C-NMR (CDCl_3_) δ (ppm): 55.5 (OCH_3_), 67.8 (CH_2_), 114.3, 117.9, 127.0, 127.9, 128.9 (CH_arom,_ =CH), 121.9, 132.1, 135.7, 137.4, 160.1(C_arom_), 182.3 (C=O) (see Appendix A). Anal. Calc. For C_17_H_14_O_3_ (M = 366.30 g/mol) %C: 76.67; %H: 5.25. Found %C: 76.68; %H: 5.20.

### 3.5. Synthesis of 4′-[(4-methoxy)phenyl]-4′,5′-dihydro-4H-spiro[chroman-3,3′-pirazol]-4-one (***4***)

Compound **3** (2.5 mmol) (0.666 g) was dissolved in anhydrous acetone (4.5 mL) to give a light yellow solution. The flask with the solution was placed in the ice bath and an ethereal solution of diazomethane (10 mmol) was added in excess. The mixture in the flask was left in the freezer. The creamy precipitate was filtered off and purified by the column chromatography and crystallized from methanol. Yield: 56% m.p: 109–111.3 °C. IR (KBr) ν(cm^−1^): 3126, 3025 (C-H aromat.), 2968, 2934 (C-H aliph.), 1684 (C=O), 1607, 1580, 1544, 1514, 1476, 1468 (C=C, N=N), 1311 (C-N), 1184 (C-O-C), 1115 (C-O). ^1^H-NMR (CDCl_3_) δ (ppm): 1.58 (3H, s, OCH_3_), 3.74 (1H, dd, *J*_AB_ = 8.4 Hz, *J*_BX_ = 8.4 Hz CH_2_), 3.78 (3H, s, CH), 4.16 (1H, d, *J*_AB_ = 12.6 Hz CH_2_), 4.55 (1H, d, *J*_AB_ = 12 Hz CH_2_), 4.90 (1H, d, *J*_AB_ = 24 Hz CH_2_), 5.13 (1H, d, *J*_AB_ = 13.39 Hz CH_2_), 6.82–7.68 (8H, m, C-H aromat) (see Appendix A). ^13^C-NMR (CDCl_3_) δ (ppm): 41.9 (CH), 55.3 (OCH_3_), 69.5 (CH_2_), 85.8 (CH_2_), 98.0, 114.3, 118.2, 121.9, 128.0, 137.0 (CH_arom_),119.5, 128.8, 159.2, 161.7 (C_arom_), 186.2 (C=O) (see Appendix A). Anal. Calc. for C_18_H_16_O_3_N_2_ (M = 308.34 g/mol) % C: 70.11; % H: 5.19; % N: 9.08. Found % C: 70.10; % H: 5.20; % N: 9.10. MS (ESI+): m/z 309.4 C_18_H_16_O_3_N_2_ [M + H]+. HRMS (ESI+): *m*/*z* [M + H]^+^ calcd. for C_18_H_17_O_3_N_2_: 309.1239; found: 309.1236 (see Appendix A).

### 3.6. Synthesis of (E)-3-benzylidene-2-phenylchroman-4-one (***5***)

Compound **5** was synthesized by the same method as compound **1** [8]. Reagents: 0.01 mol (2.243 g) 2-phenylchroman-4-one, 0.01 mol (1.062 g) benzaldehyde and 5 drops of piperidine. Yield: 85% m.p: 103–105 °C. MS (ESI+): m/z 313.5 C_22_H_16_O_2_ [M + H]+. IR (KBr) ν(cm^−1^): 3068 (C-H aromat.), 2932 (C-H aliph.), 1668 (C=O), 1604, 1577, 1506, 1492, 1472, 1461 (C=C), 1142 (C-O-C). ^1^H-NMR (DMSO-d_6_) δ (ppm): 6.30 (1H, s, C2-H), 7.69 (1H, s, =CH), 6.99–7.81 (14H, m, CH aromat.) (Appendix A). ^13^C-NMR (CDCl_3_) δ (ppm): 77.3 (C2-H), 118.7, 121.8, 127.6, 128.7, 128.9, 129.8, 129.9, 136.1, 139.5 (CH_arom_, =CH), 122.2, 132.5, 134.1, 138.1, 158.9 (C_arom_), 182.7 (C=O) (Appendix A). Anal. Calc. For C_22_H_16_O_2_ (M = 312.36 g/mol) %C: 84.59; %H: 5.16. Found %C: 84.55; %H: 5.18.

### 3.7. Synthesis of 2′-phenylchromanone-3′-spiro-3,4-phenyl-1-pyrazoline (***6***)

Compound **6** was synthesized by the same method as compound **2** [8]. Compound **5** (2.5 mmol, 0.781 g) was dissolved in anhydrous acetone (6 mL) to give a clear solution. The next steps were the same as for the synthesis of compound **2**. Yield: 78% m.p: 126.5–127 °C. MS (ESI+): m/z 355.3 C_23_H_18_O_2_N_2_ [M + H]+. IR (KBr) ν(cm^−1^): 3061, 3032 (C-H aromat.), 1697 (C=O), 1602, 1579, 1493, 1473, 1460 (C=C, N=N), 1303 (C-N), 1146 (C-O-C). ^1^H-NMR (DMSO-d_6_) δ (ppm): 4.07 (1H, dd, *J*_AB_=7.16 Hz, *J*_BX_ = 1.96 Hz CH), 4.20 (1H, dd, *J*_AB_ = 11.96 Hz, *J*_BX_ = 7.16 Hz CH_2_), 4.21 (1H, dd, *J*_AB_ = 11.95 Hz, *J*_BX_ = 1.96 Hz CH_2_), 5.86 (1H, s, C2-H), 7.10–7.69 (14, m, C-H aromat.) (Appendix A). ^13^C-NMR (CDCl_3_) δ (ppm): 55.2 (CH), 77.0 (CH), 79.5 (C2-H), 112.4, 113.9, 118.2, 121.3, 127.3, 128.0, 128.5, 129.8, 135.7, 138.9, 142.8 (CH_arom_), 100.1, 119.2, 138.9, 152.8, 159.6 (C_arom_), 185.3 (C=O) (see Appendix A). Anal. Calc. For C_23_H_18_O_2_N_2_ (M = 354.41 g/mol) %C: 77.95; %H: 5.11; %N: 7.90. Found %C: 78.80; %H: 5.20; %N: 7.70.

### 3.8. Synthesis of (E)-3-benzylidene-chroman-4-one (***7***)

Compound **7** was synthesized according to Levai and Schag [22]. Reagents used for the reaction were: 0.01 mol (1.481 g) chroman-4-one, 0.01 mol (1.062 g) benzaldehyde and five drops of piperidine. Next steps were the same as for the synthesis of compound **3**. Yield: 85% m.p: 111–112 °C. MS (ESI+): m/z 237.4 C_18_H_16_O_3_N_2_ [M + H]+. IR (KBr) ν(cm^−1^): 3056, 3028 (C-H aromat.), 2904, 2854 (C-H aliph.), 1666 (C=O), 1600, 1572, 1461, 1454 (C=C), 1306 (C-O), 1144 (C-O-C). ^1^H-NMR (DMSO-d_6_) δ (ppm): 5.43 (1H, s, =CH), 7.59 (2H, d, *J*_AB_ = 14.22 Hz C2-H), 7.05–7.90 (9H, m, C-H aromat) (see Appendix A). ^13^C- NMR (CDCl_3_ δ (ppm): 67.6 (CH_2_), 108.2, 119.3, 121.9, 125.2, 128.6, 129.5, 129.9, 131.1, 134.5 (CH_arom,_ =CH), 137.5, 154.5, 155.9 (C_arom_), 182.3 (C=O) (see Appendix A). Anal. Calc. for C_16_H_12_O_2_ (M = 236.26 g/mol) %C: 81.33; %H: 5.08. Found %C: 81.03; %H: 5.10.

### 3.9. Synthesis of 4′-phenyl-4′,5′-dihydro-4H-spiro[chromano-3,3′-pirazol]-4-one (***8***)

Compound **8** was synthesized by the same method as compound **4** [9]. Compound **7** (2.5 mmol, 0.590 g) was dissolved in anhydrous acetone (5 mL) to give a clear solution. Next step were the same as for the synthesis of compound **4**. Yield: 92.5% m.p: 141–142 °C. MS (ESI+): m/z 279.4 C_17_H_14_O_2_N_2_ [M + H]+. IR (KBr) ν(cm^−1^): 3064, 3031 (C-H, aromat.), 1677 (C=O), 1606, 1576, 1543, 1473, 1464 (C=C, N=N), 1309 (C-N), 1148 (C-O-C). ^1^H-NMR (DMSO-d_6_) δ (ppm): 4.15 (1H, dd, *J*_AB_ = 12.01 Hz, *J*_BX_ = 6.98 Hz CH_2_), 4.46 (1H, dd, *J*_AB_ = 7.58 Hz, *J*_BX_ = 6.98 Hz CH), 4.94 (1H, d, *J*_AB_ = 12.56 Hz CH), 5.12 (1H, d, *J*_AB_ = 12.56 CH), 7.09–7.68 (9, m, C-H aromat) (see Appendix A). ^13^C-NMR (CDCl_3_) δ (ppm): 42.4 (CH), 69.6 (CH_2_), 85.7 (CH_2_), 98.0, 118.2, 122.0, 128.0, 137.1, (CH_arom_), 119.5, 129.0, 161.8 (C_arom_), 186.1 (C=O) (see Appendix A). Anal. Calc. for C_17_H_14_O_2_N_2_ (M = 278.31 g/mol) %C: 73.36; %H: 5.03; %N: 10.06. Found %C: 73.40; %H: 5.13; %N: 10.10.

### 3.10. Synthesis of (E)-3-(3-methoxybenzylidene)-2-phenylchroman-4-one (***9***)

Compound **9** was synthesized by the same method as compounds **1** and **5** [8]. Reagents: 0.01 mol (2.243 g) 2-phenylchroman-4-one, 0.01 mol (1.362 g) 3-methoxybenzaldehyde and five drops of piperidine. Yield: 80% m.p.: 96.2–98.4 °C. MS (ESI+): m/z 343.5 C_23_H_18_O_3_ [M + H]+. IR (KBr) ν(cm^−1^): 3064 (C-H aromat.), 2958, 2829 (C-H aliph), 1663 (C=O), 1608, 1599, 1581 (C=C). 1176 (C-O), 1140 (C-O-C). ^1^H-NMR (DMSO-d_6_) δ (ppm): 2.28 (3H, s, OCH_3_), 6.74 (1H, s, C2-H), 7.99 (1H, s, =CH), 7.04–7.81 (13H, m, C-H aromat) (see Appendix A). ^13^C-NMR (DMSO-d_6_) δ (ppm): 54.9 (OCH_3_), 77.2 (C2-H), 115.1, 116.1, 118.8, 126.9, 127.4, 128.8, 129.0, 130.1, 134.6, 136.7, 138.9 (CH_arom_, =CH), 121.5, 122.1, 122.2, 131.8, 137.7, 158.3 (C_arom_), 181.3 (C=O) (see Appendix A). Anal. Calc. For C_23_H_18_O_3_ (M = 342.39 g/mol) %C: 80.68; %H: 5.26. Found %C: 81.60; %H: 5.55.

### 3.11. Synthesis of 5′-(3-methoxyphenyl)-2-phenyl-4′,5′-dihydro-4H-spiro[chromano-3,3′-pyrazol]-4-one (***10***)

Compound **10** was synthesized by the same method as compounds **2** and **6** [8]. Reagents used for the reaction were: 2.5 mmol (0.856 g) of compound **9** dissolved in anhydrous acetone (6 mL) to give a clear solution. Next steps were the same as for the synthesis of compound **2**. Yield: 41.3% m.p. 126–128 °C. MS (ESI+): m/z 385.4 C_24_H_20_O_3_N_2_ [M + H]+. IR (KBr) ν(cm^−1^): 3027 (C-H aromat.), 2975, 2828 (C-H aliph.), 1677 (C=O), 1603, 1579, 1550, 1515, 1472 (C=C, N=N), 1307 (C-N), 1232 (C-O), 1146 (C-O-C). ^1^H-NMR (DMSO-d_6_) δ (ppm): 2.13 (3H, s, OCH_3_), 2.31 (1H, dd, *J*_AB_ = 3.1Hz, *J*_BX_ = 6.9 Hz, CH), 6.01 (1H, s, C2-H), 6.36 (1H, dd, *J*_AB_ = 18Hz, *J*_BX_ = 3Hz CH_2_), 6.37 (1H, dd, *J*_AB_ = 18Hz, *J*_BX_ = 7.15 Hz CH_2_), 6.73–7.82 (13H, m, C-H aromat.) (Appendix A). ^13^C-NMR (DMSO-d_6_) δ (ppm): 20.6 (CH), 39.8 (OCH_3_), 81.1 (C2-H), 118.6, 121.8, 126.8, 127.8, 128.2, 128.6, 138.0 (CH_arom_) 102.2, 119.8, 134.7, 134.9, 136.1, 159.4 (C_arom_), 185.5 (C=O) (Appendix A). Anal. Calc. For C_24_H_20_O_3_N_2_ (M = 384.43 g/mol) %C: 74.98; %H: 5.20; %N: 7.34. Found %C: 74.77; %H: 5.25; %N: 7.45.

### 3.12. Determination of Lipophilicity of Flavone Analogues Using a RP-TLC Method

The RP-TLC experiments were performed on TLC plates (5 × 10 cm) RP-18 F254S (Merck, Darmstadt, Germany). The synthesized compounds were dissolved in *N*,*N*-dimethylformamide DMF (2 mg/mL). DMF was obtained from Chempur (Piekary Slaskie, Poland) and used with high purity (99.8%). The solutions of each compound in DMF were spotted on the plates, and observed under UV light at λ = 254 nm. A DMF-water solvent system was used as mobile phase. The composition of the solvent system has changed from 50%:50% to 95%:5%. All experiments were performed at room temperature. The log*P* parameter was calculated using the equation from the calibration curve.

### 3.13. Cells Cultures and Cytotoxicity Assay by MTT

Cytotoxicity was tested against human skin melanoma cells (WM-115, ECACC, Salisbury, UK), two human leukemia cell lines, *viz*. promyelocytic leukemia (HL-60) and lymphoblastic leukemia (NALM-6), and human colon adenocarcinoma cells (COLO-205). The cell lines COLO-205, WM-115 and HL-60 used in this work came from the ATCC American Type Culture Collection (Manassas, VA, USA), whereas the NALM-6 cell line was purchased from the German Collection of Microorganisms and Cell Cultures (Braunschweig, Germany). The leukaemia cells and colon adenocarcinoma were cultured in RPMI 1640 medium (Invitrogen, Grand Island, NY, USA) supplemented with 10% fetal bovine serum (FBS; Invitrogen) and gentamicin (25 µg/mL; KRKA, Novo Mesto, Slovenia). For melanoma WM-115 cells, Dulbecco’s minimal essential medium (DMEM; Invitrogen) was used. All cell lines were cultured at 37 °C in a humidified atmosphere of 5% CO_2_ in air.

For all experiments, the studied compounds were dissolved in DMSO (Sigma-Aldrich) and were further diluted in culture medium to obtain <0.1% DMSO concentration. In each experiment, controls with and without 0.1% DMSO were performed. The cytotoxicity of all compounds and of the reference compounds 4-chromanone and 3-benzylideneflavanone was determined by the MTT assay, i.e., 3-(4,5-dimethylthiazol-2-yl)-2,5-diphenyltetrazolium bromide (Sigma, St. Louis, MO, USA), which measures cellular dehydrogenase activity [24]. Exponentially growing cells were seeded a day before the experiment onto a 96-well microplates (Nunc, Roskilde, Denmark) at a density up to 6–8 × 10^3^ cells/well (depending on the cell line). Subsequently, various concentrations of the studied compounds freshly prepared in DMSO and diluted with complete culture medium were added. All compounds were tested for their cytotoxicity at a final concentration of 10^−7^–10^−3^ M. After 46 h of incubation with the studied compounds, the cells were treated with the MTT reagent and incubation was continued for another two hours. MTT – formazan crystals were dissolved in 20% sodium dodecyl sulphate (SDS, Sigma-Aldrich) and 50% DMF (Sigma-Aldrich) at pH 4.7; following this, absorbance was read at 570 nm on a multifunctional Victor ELISA-plate reader (Perkin Elmer, Turku, Finland). The IC_50_ values, i.e., the concentration of the test compound required to reduce the cell survival fraction to 50% of controls, were calculated from concentration-response curves and used as a measure of the sensitivity of the cells to a given treatment. As a control, cultured cells were grown in the absence of drugs. The data points represent the means of at least five to ten repeats ± standard deviation (S.D.).

### 3.14. Red Blood Cells Lysis Assay

The studies on biological material were approved by the Bioethics Committee of the Medical University of Lodz, Poland (RNN/109/16/KE). The blood samples were obtained from healthy donors at the Blood Donation Centre in Lodz. The procedure for plasma preparation for erythrotoxicity studies was described previously [10]. The influence of synthesized compounds as well as two reference compounds flavanone and chromanone on RBC membrane integrity was performed by lysis assay which was conducted according to our previous protocol [10]. Briefly, 2% RBC suspension in 0.9% NaCl was incubated at 37 °C for one hour with the tested compounds at various concentrations corresponding to their activity towards cancer cell lines or 0.9% NaCl (control). The samples were centrifuged at 3000 rpm for 10 min and the absorbance of the supernatant was measured at 550 nm. The results are presented as percentage of haemolysis where a sample containing 10 μL of 2.0% *v/v* Triton X-100 was regarded as a positive control contributing to 100% of haemolysis. A sample containing saline solution represented spontaneous haemolysis of RBCs (control). The studies were performed on four biological samples obtained from different blood donors (n = 4). The results are presented as mean ± standard deviation (SD). The coefficient of variability for the method was counted: W = 9.51%, *n* = 5.

### 3.15. RBCs Morphology

A 2% erythrocyte suspension was incubated at 37 °C for one hour with various concentrations of synthesized compounds. The morphology of the RBCs was examined by means of a phase contrast Opta-Tech inverted microscope, at 400× magnification, equipped with software (OptaView 7) for image analysis.

### 3.16. Cell Cycle Analysis

One million HL-60 (subline CCL 240; cell type: promyeloblast) cells in 3 mL of RPMI 1640 supplemented with 20% FBS were seeded on 6-well plate. Following 20 h incubation with the test compounds at the concentrations of 0.5 × IC_50_ and 1 × IC_50_, or 1% DMSO as the vehicle control, the cells were centrifuged (6 min, 700 rpm, room temperature (RT)), washed with PBS (without Ca^2+^ and Mg^2+)^ and suspended in 1 mL of cold 70% ethanol in PBS. Fixed cells were stored in −20 °C for 2–3 days. Before analysis, cells were centrifuged (10 min, 1500 rpm, 4 °C) and washed twice with cold PBS. After the second wash, the cells were suspended in 1mL of PBS containing 100 μg of RNase A and incubated in 37 °C for 10 min. Next, 5 μL of solution of propidium iodide (1 mg/mL) was added to each sample and incubated in dark for 30 min at RT. Stained cells were analyzed by flow cytometry (FACS Calibur, BD Biosciences, San Jose, CA, USA). Data from 10,000 gated cells were collected using CellQuestPro software (Ver 6.0, Becton Dickinson) and cell cycle analysis was performed with ModFit LT software (Ver 3.2, Verity Software House, Topsham, ME, USA).

### 3.17. Digestion of Plasmid DNA with BamHI Restriction Nuclease

First, 0.5 µg of plasmid DNA (pcDNAHisC, total length 5.5 kbp) containing a unique BamHI restriction site was dissolved in a 1x BamHI reaction buffer and incubated overnight at 37 °C with the test compounds or daunorubicin, which was used as a positive control. The concentration of the test compounds and daunorubicin in the samples was 10 µM. The final concentration of DMSO in all samples was 10%. In the next step, the reaction mixtures were digested with BamHI restriction endonuclease (2 U/µL) for 3 h at 37 °C. Total reaction volume was 10 µL. Products of the reaction were separated in the 1% agarose gel in TBE buffer. The gel was stained with ethidium bromide and DNA fragments were visualized under a UV lamp (GBox, Syngene, Frederick, MD, USA).

### 3.18. X-ray Diffraction Experiment

X-ray data for compounds **2** and **8** were collected on a SuperNova diffractometer (Agilent, Santa Clara, CA, USA) at T = 100(2)K. The intensities were recorded with an Atlas detector with MoKα radiation (λ = 0.71073 Å). For all data, multi-scan absorption was applied [25]. Both structures were solved using direct methods and the least-square refinement on F^2^. All non-hydrogen atoms were refined anisotropically. SHELXT [26] and SHELXL-2014/7 programs were used to solve and refine the structures [27]. All non-hydrogen atoms were refined anisotropically. The positions of all H-atoms were calculated based on known geometry; the respective C-H bond lengths for aromatic CH, methine CH and methyl CH_3_ atoms are 0.95, 1.00 and 0.98 Å. The H atoms were included as riding contributions, with isotropic thermal parameters set at 1.5 and 1.2 times the Ueq of the parent atom (both the methyl and remaining atoms). To identify molecular geometries and hydrogen-bond patterns, PLATON [28] and MERCURY [29] were used. The experimental, refinement and crystallographic details are shown in Table 5. Further crystallographic details for the structures reported in this paper may be obtained free of charge on application to CCDC, 12 Union Road, Cambridge CG21, EZ, UK [fax: (44) 1223-336-033; e-mail: deposit@ccdc.cam.ac.uk] on quoting the depository numbers: CCDC 1957835 for 2 and CCDC 1957892 for 8.

### 3.19. Statistical Analysis

Statistical analysis was conducted with a commercially-available package (GraphPad Prism 5, GraphPad, San Diego, CA USA). All results are presented as means ± standard deviation (SD). Normal distribution of continuous variables was confirmed with Shapiro-Wilk test, and homogeneity of variances was checked using Levene’s test. The results were analyzed using one way ANOVA, and Dunnett’s post hoc test. The results were considered significant at *p*-values lower than 0.05.

## 4. Discussion and Conclusions

This paper explores the biological activity of a series of ten compounds comprising various 3-benzylideneflavanones and 3-benzylidenechromanones, as well as their spiro- analogues with various substituents at the C3 position. The tested compounds exhibited relatively high cytotoxicity against four cancer cell lines: HL-60, NALM-6, WM-115 and COLO-205. We indicate that parent compounds **1** and **9** with a phenyl ring at C2 show lower cytotoxicity for the studied cancer cell lines than their spiropyrazolines analogues. Regarding the compounds with a methoxy substituent, compound **1** demonstrated lower cytotoxic activity than **3**. Similarly **5** was found to display lower cytotoxic activity than **7**. The most cytotoxic compound was **2**, which demonstrated an IC_50_ = 3.0 ± 0.3 µM against HL-60.

The presence of a fused pyrazoline ring may well play a key role in selective cytotoxicity. The *p*-methoxy-substituted analogues of 3-arylideneflavanone exhibit high cytotoxicity against four cancer cell lines [30,31]. The unsubstituted compound **6**, with a phenyl ring bound to the pyrazoline ring, exhibited more than 330-fold lower cytotoxicity against HL-60 than compound **2**, bearing a *p*-methoxy group on the phenyl ring.

Cytotoxicity was also strongly influenced by the position of any substituents in the phenyl ring of the spiroflavanones or spirochromanones, especially against the HL-60 cell line. Both **6** and **10** have a methoxy group in the phenyl ring, and *m*-methoxy- **10** and *p*-methoxy- **6** exhibited high activity towards the HL-60, NALM-6 and WM-115 cell lines.

One of the aims of this study is to evaluate the log*P* of the synthesized compounds. Unfortunately, no significant relationship was observed between chemical structure and lipophilicity, nor between cytotoxicity and the electron-attracting or electron-donating properties of the substituents. Some of the analyzed complexes demonstrated promising anti-proliferative properties against specific human cancer cell lines. It is worth noting that the ability to inhibit the proliferation of cancer cells was determined by the structure of the studied compounds. Additional structure–activity relationship (SAR) studies are needed to better understand the relationship between the structure of the compound and its biological activity.

Our present findings indicate that the ability of the studied compounds to inhibit the growth of cancer cells may result from cell cycle inhibition. The compounds were found to block the cell cycle of several cancer cells in the G2/M phase. In addition, as indicated by their ability to prevent digestion of plasmid DNA with BamHI restriction nuclease, the studied chromanone/flavanone analogues appear to be capable of intercalating with genomic DNA of cancer cells, thus leading to cell damage and ultimately apoptosis. It is noteworthy that commercially-used anticancer drugs, such as docetaxel, doxorubicin, cisplatin or etoposide, are also known to arrest the cycle of cancer cells in the G2/M phase [32,33,34]. Our findings correspond closely with those of previous reports on the cell cycle distribution in HL-60 cells following treatment with cisplatin. Velma et al. [35] demonstrated that cisplatin caused changes in the expression of genes involved in cell cycle regulation and programmed cell death among HL-60 cells, with both cell cycle blockade and apoptotic pathway activation being observed. It is worth noting that not only do our chromanone/flavanone analogues demonstrate similar molecular anti-proliferative activity to cisplatin, but they also possess the ability to interact with DNA and damage it [35]. Our results are in agreement with those of previous experiments carried out on leukemia cells exposed to etoposide, another commonly-used chemotherapeutic drug. Etoposide has also been shown to inhibit the cell cycle of HL-60 cells in the G2/M phase [35].

Our findings also indicate that the test compounds protected DNA against cleavage catalyzed by BamHI endonuclease in a similar way to quercetin, which has also previously been shown to intercalate with DNA [36]. This suggests that the test compounds interact with DNA, possibly by intercalation. Furthermore, the test compounds arrested leukemia cells in the G2/M phase to an extent comparable with quercetin, suggesting that they share a similar mechanism of action.

We intend to continue this line of research into chromanone/flavanone analogues towards better understanding their influence on the mechanisms of cell cycle arrest of cancer cells. It is planned to extend our research to include expression analysis of genes involved in cell cycle regulation and apoptosis.

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
