# Peer review of "Biological Evaluation of 3-Benzylidenechromanones and Their Spiropyrazolines-Based Analogues"

_molecules, 2020, doi:10.3390/molecules25071613_

Round 1

Reviewer 1 Report

The authors have significantly improved the material of the article. The material has become more understandable to the reader. The work makes a favorable impression. However, there are a few comments on the article.

On page 2 line 80, the authors write (The compounds were tested for biological activity on five cell lines: HL-60 (human leukemia cell line) , NALM-6 (human peripheral blood leukemia cell line), WM-115 (melanoma cell line) and COLO-205 (human colon adenocarcinoma cells).). However, there are only four lines of cancer cells. Maybe should be …activity on four cell lines…?

Also on page 2 line 84, the authors write (In the previous study [11], and the present one, the compounds were tested for biological activity on four cell lines: HL-60, NALM-6 and WM-115). Maybe should be …activity on three cell lines…?

The text block lines 79-81 and the text block lines 84-86 are duplicated. Please correct this paragraph.

The authors should use the ACS style for all schemes. Schemes should be friendly for the reader.

Also detailed description of the spectra of compounds in Supporting information should be friendly for the reader. Please improve this part.

On page 5 lines 156-160 the authors explain the high activity of the compound A1a in that this compound contains a bulky phenyl substituent is attached in the C2 position. However, in the case of compounds A1a and A3a the activity is deference, for example, IC 50 for COLO-205 is 29.3vs487.1. The structures of this compounds almost the same, only one the difference is position of OCH3 group in phenyl fragment (p-OMe vs m-OMe). How does this small difference affect activity?

That the authors can say about the toxicity of tested compounds? What about the selectivity index? SI = IC50(normal cells)/IC50(cancer cells)? Maybe the tested compound are toxic for normal cells. The authors should add the SI values for tested compounds.

What does a star mean near IC50 values for compound A1a?

Grams and moles should be indicated or all compounds in the experimental part.

All graphical NMR spectra should be presented in accordance with the rules of the Molecules journal (. Physical and Spectroscopic Data: Physical and spectroscopic data as well as tables for NMR data should be prepared according to the ACS's Preparation and Submission of Manuscripts standard) - See Instructions for Authors. There is a link to improve the graphical spectra. https://publish.acs.org/publish/author_guidelines?coden=joceah (Data Requirements – NMR).

Also, please read the HRMS Spectrum Description Rules (HRMS (ESI/Q-TOF) m/z: [M + Na]+ Calcd for C13H17NO3Na 258.1101; Found 258.1074. The number of potential molecular formulas within a given mass range centered on a measured (Found) value increases rapidly with molecular mass. A Found value within 0.003 m/z unit of the Calcd value of a parent-derived ion, together with other available data (including knowledge of the elements present in reactants and reagents) is usually adequate for supporting a molecular formula for compounds with molecular masses below 1000 amu (see comment above).

Please provide HRMS Spectrum description with m/z to the fourth decimal place. Also please provide graphical HRMS spectra in Supporting information.

After corrections and addition of selectivity index (SI) values for the tested compounds, the article can be accepted for publication in the journal Molecules.

Author Response

Response to the Reviewer 1

  1. Comment: On page 2 line 80, the authors write (The compounds were tested for biological activity on five cell lines: HL-60 (human leukemia cell line) , NALM-6 (human peripheral blood leukemia cell line), WM-115 (melanoma cell line) and COLO-205 (human colon adenocarcinoma cells).). However, there are only four lines of cancer cells. Maybe should be …activity on four cell lines…?

Response: The authors thank to the Reviewer for the comments. In the manuscript on the page 2 the authors made a mistake. We conducted the biological activity on four cell lines, not five cell lines. We corrected the mistake in the main text.

  1. Comment: Also on page 2 line 84, the authors write (In the previous study [11], and the present one, the compounds were tested for biological activity on four cell lines: HL-60, NALM-6 and WM-115). Maybe should be …activity on three cell lines…?

Response: The authors mistakenly wrote “four cell lines”, there should be “three cell lines”.

  1. Comment: The text block lines 79-81 and the text block lines 84-86 are duplicated. Please correct this paragraph.

Response: The authors corrected the paragraph and removed the  duplicate sentence.

  1. Comment: The authors should use the ACS style for all schemes. Schemes should be friendly for the reader.

Response: The authors corrected the schemes in the manuscript and used ChemDraw programme.

  1. Comment: Also detailed description of the spectra of compounds in Supporting information should be friendly for the reader. Please improve this part.

Response: The authors improved the spectra of the compounds and included  in Supporting Information.

  1. Comment: On page 5 lines 156-160 the authors explain the high activity of the compound A1a in that this compound contains a bulky phenyl substituent is attached in the C2 position. However, in the case of compounds A1a and A3a the activity is deference, for example, IC 50 for COLO-205 is 29.3vs487.1. The structures of this compounds almost the same, only one the difference is position of OCH3 group in phenyl fragment (p-OMe vs m-OMe). How does this small difference affect activity?

Response: The further analysis was done and appropriate fragment was added to the manuscript (page 5, lines: 158 – 163). However, we are aware to draw final conclusions a more systematic analysis is needed. It would be also necessary to have all crystal structures (it is not possible because of bad quality not crystalline compound). What we have included in the manuscript should be treated as the observation, not general conclusions.

  1. Comment: That the authors can say about the toxicity of tested compounds? What about the selectivity index? SI = IC50(normal cells)/IC50(cancer cells)? Maybe the tested compound are toxic for normal cells. The authors should add the SI values for tested compounds.

Response: First of all, thank you for your valuable attention regarding the cytotoxicity of tested compounds in relation to normal cells and the selectivity index. We are aware that the selectivity of available cancer drugs and potential chemotherapeutics is currently one of the key problems of modern medicine. In our preliminary research, we focused only on the anti-tumor properties of the tested compounds, their cytotoxicity, the effect on the cell cycle of specific tumor cell lines, and the impact on erythrocyte morphology. In the next stage, we plan to significantly expand research on the molecular mechanism of action of these compounds. One of the first and key stages will be to examine the cytotoxic properties of the tested compounds against normal endothelial cells - HUVEC cell lines isolated by us from the umbilical vein and HMEC-1 and determining the selectivity index. Low or no cytotoxicity to human normal endothelial cells will be a prerequisite for further testing. We plan to examine the proapoptotic properties of the analyzed derivatives, the ability to trigger autophagy or the effect on changes in the expression of genes involved in the proliferation of cancer cells.

  1. Comment: What does a star mean near IC50 values for compound A1a?

Response: The compound A1a was marked with a star by mistake. Only the reference compounds are marked with a star. We removed the „star” near the IC50 values for A1a.

  1. Comment: Grams and moles should be indicated or all compounds in the experimental part.

Response: The authors completed the missing units (grams and moles) for all compounds described in the experimental part.

  1. Comment: All graphical NMR spectra should be presented in accordance with the rules of the Molecules journal (. Physical and Spectroscopic Data: Physical and spectroscopic data as well as tables for NMR data should be prepared according to the ACS's Preparation and Submission of Manuscripts standard) - See Instructions for Authors. There is a link to improve the graphical spectra. https://publish.acs.org/publish/author_guidelines?coden=joceah (Data Requirements – NMR).

Response: The spectra of compounds were processed again in MestRe-C programme and included in Supporting Information. In the manuscript (Materials and methods) we wrote the shifts for deuterated solvents (DMSO-d6 and CDCl3) in the 1H and 13C NMR.

  1. Comment: Also, please read the HRMS Spectrum Description Rules (HRMS (ESI/Q-TOF) m/z: [M + Na]+ Calcd for C13H17NO3Na 258.1101; Found 258.1074. The number of potential molecular formulas within a given mass range centered on a measured (Found) value increases rapidly with molecular mass. A Found value within 0.003 m/z unit of the Calcd value of a parent-derived ion, together with other available data (including knowledge of the elements present in reactants and reagents) is usually adequate for supporting a molecular formula for compounds with molecular masses below 1000 amu (see comment above).

Please provide HRMS Spectrum description with m/z to the fourth decimal place. Also please provide graphical HRMS spectra in Supporting information.

Response: Due to the high cost of HRMS spectra and long waiting times, only two compounds were selected. HRMS spectra were made for compounds with the highest cytotoxic activity among spiro- compounds (2, 4). The graphical HRMS spectra were included in Supporting Information (Figure S5b and Figure S11b).

Reviewer 2 Report

Review of Ms. Molecules-728173 entitled “Biological evaluation of 3-benzylidenechromanone and their pyrazolines derivativesby A. A. Adamus-Grabicka et al.

The authors have successfully addressed the majority of my concerns. Even though, there are still some issues to be addressed by the authors in order to find the manuscript suitable to be published in Molecules.

Some major concerns are found regarding cell cycle arrest studies. The authors claim that they performed these studies after 20h of incubation due to cell death but this should not be a problem in standard conditions for such studies (working at IC25 and IC0 values).   The authors should have into account the cell cycle duration in HL-60 in order to assure that at least the time needed for completing a cell cycle have been apllied for incubation with the compounds under study. The author should include in the experimental part, the HL-60 subline employed (include ATCC code) since the cell cycle duration varies from 24 to 48 h in this cell line. The authors should also specify  the percentage of DMSO in cell cycle studies. From the results, it is clear that in this case, the vehicle (DMSO) has a crystal clear effect in cell cycle progression. Therefore, the conclusions extracted for the compounds cannot be well supported. Indeed, it should be noted that DMSO was reported as a chemical to be used for differentiation purposes in this cancer cell line (https://www.lgcstandards-atcc.org/Products/All/CCL-240.aspx?geo_country=es#characteristics) with the effects underlying this differenciation process  (Birnie GD. The HL60 cell line: a model system for studying human myeloid cell differentiation. Br J Cancer Suppl. 1988;9:41–45)

Finally, I have some major concerns in plasmid digestions experiments. The authors should improve the given details in the experimental part. 100 µM of the studied compounds is a high concentration. They are in a pretty excess respecting the plasmid.  With such an excess DNA intercalation could not be the cause of inhibition cleavage since such an amount of drug molecules can be aggregated for instance inhibiting plasmid recognition by BamH1. On the other hand, BamH1 interaction, poisoning, should be considered. Indeed not all the reported drugs able to inhibit plasmid digestion are DNA intercalating agents. What is the % of DMSO employed? From my experience, DMSO does not affect plasmid migration but it may affect enzyme activity. The authors should include a vehicle control in this experiment.

Minor comments:

  • Line 80: replace five cell lines by four cell lines
  • Line 83: with DNA and cell cycle arrest studies
  • From line 80 to 90: please rewrite, it is repetitive.
  • I strongly recommend the author to include Figure 6 with the chemical structure of A7 and B6 in Scheme 1 indicating that they have been reported in a previous work. It would be easier for readerships to follow the manuscript.
  • In the same way, please insert A7 and B6 data regarding IC50 and LogP values in Table 3 with a table note indicating that they were taken from reference 10 to facilitate comparisons.
  • Histograms in Fig. 8 can be moved to SI.

Author Response

Response to the Reviewer 2

  1. Comment: Some major concerns are found regarding cell cycle arrest studies. The authors claim that they performed these studies after 20h of incubation due to cell death but this should not be a problem in standard conditions for such studies (working at IC25 and IC0 values).   The authors should have into account the cell cycle duration in HL-60 in order to assure that at least the time needed for completing a cell cycle have been apllied for incubation with the compounds under study. The author should include in the experimental part, the HL-60 subline employed (include ATCC code) since the cell cycle duration varies from 24 to 48 h in this cell line.

Response: The author included in the manuscript the subline  of HL-60 used in the experiments. The used subline of HL-60 was CCL-240 collected from a patient with acute promyelocytic leukemia.

HL-60 ATCC CCL-240 (the doubling time for this cell line is 24 – 28h https://dtp.cancer.gov/). We used asynchronous cells which means that we have a mixture of cells in different cell cycle phases. As shown in Fig 8 (manuscript) more than 50% of HL-60 cells are in S and G2M phase. This means that they need much less time than 24-28 h to complete the cell cycle. Therefore 20 h of incubation with test compounds is long enough for measuring effects on the cell cycle arrest. Tang et al. (Cancer Lett. 2011 November 1; 310(1): 15–24) showed that cell cycle arrest in HL-60 cells can be measured as soon as 8, 16 and 24 h after exposure to cytotoxic agent. Longer exposure times lead to increase of cells in sub G1 phase indicating the presence of dead (or fragmented) cells. Cabrera et al. ((2015) PLoS ONE 10(9): e0136878. doi:10.1371/journal.pone.0136878) studied HL-60 cell cycle arrest after 15 h of incubation with toxic compound.

  1. Comment: The authors should also specify  the percentage of DMSO in cell cycle studies. From the results, it is clear that in this case, the vehicle (DMSO) has a crystal clear effect in cell cycle progression. Therefore, the conclusions extracted for the compounds cannot be well supported. Indeed, it should be noted that DMSO was reported as a chemical to be used for differentiation purposes in this cancer cell line (https://www.lgcstandards-atcc.org/Products/All/CCL-240.aspx?geo_country=es#characteristics) with the effects underlying this differenciation process  (Birnie GD. The HL60 cell line: a model system for studying human myeloid cell differentiation. Br J Cancer Suppl. 1988;9:41–45)

Response: The concentration of DMSO in the cell cycle studies was 1 %. The manuscript was corrected in the Materials and Methods section and in the description of Fig 8. Indeed, 1% DMSO can induce differentiation of HL-60 cells to granulocytes but incubations longer than 20 h are required. For example, Tarella et al. (Cancer Research 1982; 42:445-449) demonstrated that 30 h exposure of HL-60 cells to 1.2 % DMSO has no effect on proliferation and functional or morphological differentiation of HL-60 cells.

    We agree with the Reviewer that in our experimental settings 1% DMSO (vehicle control) increased the number of cells in G1 phase when compared to non-treated HL-60 cells (manuscript Fig 8 b). However, test compounds 11, 12 or quercetin showed obviously different effect and caused accumulation of HL-60 cells in G2/M phase. Note, that in the presence of DMSO less than 10% of cells are in G2/M, while in the presence of 11 at least 60% of cells are accumulated in G2/M phase. Therefore, we think our conclusions that test compounds cause accumulation of HL-60 cells in G2/M phase are valid and well supported by experimental data.

  1. Comment: Finally, I have some major concerns in plasmid digestions experiments. The authors should improve the given details in the experimental part. 100 µM of the studied compounds is a high concentration. They are in a pretty excess respecting the plasmid.  With such an excess DNA intercalation could not be the cause of inhibition cleavage since such an amount of drug molecules can be aggregated for instance inhibiting plasmid recognition by BamH1. On the other hand, BamH1 interaction, poisoning, should be considered. Indeed not all the reported drugs able to inhibit plasmid digestion are DNA intercalating agents. What is the % of DMSO employed? From my experience, DMSO does not affect plasmid migration but it may affect enzyme activity. The authors should include a vehicle control in this experiment.

Response: We have repeated plasmid digestion experiments using test compounds at the concentration of 10 µM. We obtained similar results as observed previously. This strongly supports our conclusion that test compounds intercalate to DNA. Figure 7 was replaced with the new data. The concentration of DMSO in all samples was 10 %. Our results indicate that 10 % DMSO has no effect on BamHI activity because the plasmid DNA was fully linearized in the vehicle control (Fig 8, lane B). Materials and Methods section and description of Fig 8 regarding DMSO concentration and the concentration of the test compounds were corrected.

  1. Comment: Line 80: replace five cell lines by four cell lines

Response: The authors made a mistake of entering five cell lines in the manuscript. In the manuscript should be 4 cell lines. The authors corrected the mistake in the main text.

  1. Comment: Line 83: with DNA and cell cycle arrest studies. From line 80 to 90: please rewrite, it is repetitive.

Response: The authors thank to the reviewer for pointing out the mistakes in the manuscript. We corrected the sentence in the line 83 and rewrite the part of the text from line 80 to 90.

  1. Comment: I strongly recommend the author to include Figure 6 with the chemical structure of A7 and B6 in Scheme 1 indicating that they have been reported in a previous work. It would be easier for readerships to follow the manuscript. In the same way, please insert A7 and B6 data regarding IC50 and LogP values in Table 3 with a table note indicating that they were taken from reference 10 to facilitate comparisons.

Response: The authors corrected the Scheme 1 and included the chemical structure of the compounds A7 and B6. In the Table 3 we added the IC50 and logP values for compounds A7 and B6 too. In addition, we changed the numbering of the compounds

classic (1, 2, 3, 4, 5, 6, 7, 8, 9, 10, 11, 12) as suggested the Reviewer 3.

  1. Comment: Histograms in Fig. 8 can be moved to SI

Response: The authors moved the histograms of DNA content in HL-60 cells treated with compounds at a concentration of 1xIC50 to the Supporting Information.

Reviewer 3 Report

  1. Budzisz describe the synthesis and characterization of some chromanones, including interesting results from bioassays. ORTEPs and analysis to confirm unequivocally the structures of a couple tested compounds are provided. Lipophilicity behind oral bioavailability was measured, good values were found, being in line with Lipinski rules. A considerable number of tests were performed to evaluate their cytotoxicity toward a panel of human carcinoma cell lines (HL-60, NALM-6, WM-115 and COLO-205). Erythrocyte safety was evaluated. Thus, I found enough originality and a lot of work worthy of publication in Molecules, but after fixing some minor points, as follows:
  • The title should emphasize what kind of biological activity was evaluated. Thus, a better one could be ‘Cytotoxicity evaluation of 3-benzylidenechromanones and their spiropyrazoline-based analogues’
  • Chemical structure drawings (Scheme 1) must be corrected using the template of ‘Molecules’ in ChemDraw. Compare, for example, the pyrazole with a phenyl ring from the products, the former seems to be constrained. In the same context, Authors stated in Ln 117 – Pg.3 that the products were synthesized in a regioselective and stereospecific manner, but, the drawings do not help to visualize that issue.
  • I strongly recommend to use classic numbering for structures instead of the current (A1, A2a,..)
  • The abstract is too large. Thus, it should be written in concise form. In line, conclusion part could also be decreased.
  • The use of the term ‘derivative’ should be revised throughout the manuscript because it is misinterpreted and overused. ‘Derivative’ means that something A (derivative of B or B derivative) just comes from something B. In this context, authors should consider using the term ‘analogues’, where applicable, e.g. in the title.
  • Most parts of the manuscript are fully detailed. However, it would have been good to embed the reaction mechanism, for instance, after Ln 77-Pg 1. It may be useful for readers from synthetic community
  • Ln 98 – Pg 3. And conclusions, manuscripts cannot examine things
  • In the authors contribution, how authors did calculate those participation rates?
  • With respect to the Supporting Information File: format for spectra must be homogenized, some 1H NMR spectrums are embedded into a dark blue canvas (difficult to print). For all 1H NMR spectra, at least integration, coupling constants and peaking must be visible. The Figure S2, excceds the page format, is so large.  After Figure S7, the grid for those spectrums is undesirable.  

Author Response

Response to the Reviewer 3

  1. Comment: The title should emphasize what kind of biological activity was evaluated. Thus, a better one could be ‘Cytotoxicity evaluation of 3-benzylidenechromanones and their spiropyrazoline-based analogues’

Response: The authors are very grateful for the comments on the manuscript. We decided to change the title of the article on: Biological evaluation of 3-benzylidenechromanone and their spiropyrazolines-based analogues. Previously the title of the manuscript was changed because the Reviewers concluded that the research concerned not only cytotoxicity tests but biological studies in general.

  1. Comment: Chemical structure drawings (Scheme 1) must be corrected using the template of ‘Molecules’ in ChemDraw. Compare, for example, the pyrazole with a phenyl ring from the products, the former seems to be constrained. In the same context, Authors stated in Ln 117 – Pg.3 that the products were synthesized in a regioselective and stereospecific manner, but, the drawings do not help to visualize that issue.

Response: The Scheme 1 was corrected using ChemDraw programme. The authors prepared a scheme to help to better understand presented the regioselective and stereospecific manner.

  1. Comment: I strongly recommend to use classic numbering for structures instead of the current (A1, A2a,..)

     Response: The authors changed the current numbering of the compounds on the classic one (1, 2, 3, 4, 5, 6, 7, 8, 9, 10, 11, 12).

  1. Comment: The abstract is too large. Thus, it should be written in concise form.

Response: The authors corrected and decreased the abstract.

  1. Comment: The use of the term ‘derivative’ should be revised throughout the manuscript because it is misinterpreted and overused. ‘Derivative’ means that something A (derivative of B or B derivative) just comes from something B. In this context, authors should consider using the term ‘analogues’, where applicable, e.g. in the title.

Response: We considered to use of the “derivative” term and decided to change it on “analogue” term in some sentences.

  1. Comment: Most parts of the manuscript are fully detailed. However, it would have been good to embed the reaction mechanism, for instance, after Ln 77-Pg 1. It may be useful for readers from synthetic community.

     Response: The authors presented in the manuscript the formation of the spiro- compounds to mechanism of the reaction.

  1. Comment: Ln 98 – Pg 3. And conclusions, manuscripts cannot examine things

Response: The authors agree the term "examine things" was used incorrectly. We deleted it from the main manuscript.

  1. Comment: In the authors contribution, how authors did calculate those participation rates?

     Response: The participation rates were calculated based on co-authors statement made the part of the studies.

  1. Comment: With respect to the Supporting Information File: format for spectra must be homogenized, some 1H NMR spectrums are embedded into a dark blue canvas (difficult to print). For all 1H NMR spectra, at least integration, coupling constants and peaking must be visible. The Figure S2, excceds the page format, is so large.  After Figure S7, the grid for those spectrums is undesirable.  

Response: The authors improved the spectra of the compounds and included them in supporting  information. The figure S2 was fitted to the page format. The grid in the spectra has been removed.

Reviewer 4 Report

I would not recommend publishing this article in Molecules until the following issues are addressed:

1) There are several grammatical mistakes throughout

2) The abstract and introductory paragraph refer to complex chemical structures and reactions, yet there are no figures or schemes showing the structure of these compounds or the reactions described. Please include

3) the one synthetic scheme that the paper does have, does not adequately show the stereochemistry if indeed it is stereoselective

4) There are several figures that are mislabeled and several without labels which makes following the text very difficult.

5) the paper starts with referring to Table 3, but there is not a Table 1 or 2??

6) The cytotoxicity data in Table 3 includes several compounds that do not have a chemical structure (e.g. Quercetin?)

Author Response

Response to the Reviewer 4

  1. Comment: There are several grammatical mistakes throughout

Response: The authors thank to the Reviewer for the comments to the manuscript. The English was corrected by native speaker  working at Medical University of Lodz.

  1. Comment: The abstract and introductory paragraph refer to complex chemical structures and reactions, yet there are no figures or schemes showing the structure of these compounds or the reactions described. Please include. The one synthetic scheme that the paper does have, does not adequately show the stereochemistry if indeed it is stereoselective.

Response: The authors prepared a new scheme of the reaction (Scheme 1) showing a formation a spiro- compounds. Below we included regioselective reaction 3-benzylideneflavanone and diazomethane of formation spiro- compounds.

The 1,3-dipolar cycloaddition of diazomethane to (E)- and (Z)-isomers of exocyclic α,βunsaturated ketones was found to be regioselective and stereospecific providing spiro-1-pyrazolines. The methylene moiety of the diazomethane was connected to the β-carbon atom of the α,β-enone and the stereochemistry of the starting α,β-unsaturated ketone was retained in each case.

  1. Comment: There are several figures that are mislabeled and several without labels which makes following the text very difficult.

Response: The authors introduced the missing labels and corrected mislabeled figures.

  1. Comment: The paper starts with referring to Table 3, but there is not a Table 1 or 2?

Response: The authors made a mistake of entering the Table 3 first. We have improved the order in which tables appear in the manuscript.

  1. Comment: The cytotoxicity data in Table 3 includes several compounds that do not have achemical structure (e.g. Quercetin?)

Response: The authors included a chemical structure of reference compounds.

Round 2

Reviewer 1 Report

The authors have made effort to improve paper.

However, the authors should take into account the following comments:

In the introduction, authors should provide formulas for promising pyrazolines and their analogues from the literature. Without structures, the introduction is difficult for the reader.

The authors should check Scheme 1. Here the carbon atom has V-valency. Also, this scheme should be drawn in ACS format.  

On page 7 line 199. The phrase “antitumor activity” in the context of this paper is no correct. I think, “anticancer activity” will be more correct.

The authors have improved the graphic NMR spectra, but this is not enough, because multiplicity cannot be seen.

The authors did not provide selectivity index for tested compounds.  As a control normal liver cells or normal lymphocyte cells can be used for example. The compound can be friendly to erythrocyte, but toxic to normal cells.

After correction the paper can be accepted.

Author Response

Responses to the Reviewer 1

All responses are attacht below

Reviewer 2 Report

I consider that the authors have made a great effort to address all my concerns and the manuscript have been substantially improved from a biological point of view. Therefore, I recommend publication of this manuscript in Molecules.

Author Response

Response to the Reviewer 2

  1. Comment: I consider that the authors have made a great effort to address all my concerns and the manuscript have been substantially improved from a biological point of view. Therefore, I recommend publication of this manuscript in Molecules.

Response: The authors thank the Reviewer very much for valuable suggestions and comments regarding the manuscript. We are glad we have met the Reviewer's requirements.

Reviewer 4 Report

The paper entitled Biological evaluation of 3-benzylidenechromanone and their spiropyrazoline based analogues” outlines the evaluation of a series of compounds based on flavanones. This series demonstrates some moderate activity against a variety of cancer cell lines. I would not advocate publishing this paper until the following concerns are met:

  1. Please provide more bond line structures using chemdraw or some other drawing program when discussing the various series of compounds in the introduction. For example, spiropyrazole analogs that are NOS inhibitors or CB1 antagonists.
  2. Please provide bond line structures when discussing the previous synthesis of pyrazolines
  3. Please provide more bond line structures using chemdraw or some other drawing program when discussing the synthetic scheme. The structures in Scheme 1 are very difficult to interpret. There is a pentavalent carbon in one of the structures and the authors attempt to show the 3D stereochemistry is not conventional and very difficult to understand.
  4. Please provide more bond line structures using chemdraw or some other drawing program for each compound synthesized in the table.
  5. Please provide more bond line structures using chemdraw or some other drawing program next to each x-ray structure.

Author Response

Response to the Reviewer 4

  1. Comment: Please provide more bond line structures using chemdraw or some other drawing program when discussing the various series of compounds in the introduction. For example, spiropyrazole analogs that are NOS inhibitors or CB1 antagonists.

Response: The authors thank to the Reviewer for the comments. We prepared and added structures of pyrazoline compounds and their analogues (Figure 1) which demonstrate a biological activities for example: anticancer, antifungal, antibacterial, anti-inflammatory, antidepressant, nitric oxide synthase inhibitor, cannabinoid antagonist. We have added a fragment about the structure of pyrazolines and their chemical structures depending on the location of the double bond: A – Δ1-pyrazoline, B – Δ2-pyrazoline, C – Δ3-pyrazoline.

  1. Comment: Please provide bond line structures when discussing the previous synthesis of pyrazolines.

Response: In the previous paper [10] we synthesized two compounds:

(E)-3-(4-N,N-diethylaminobenzylidene)-2-phenyl chroman-4-one (11) and (E)-3-(4-N,N-dimethylaminobenzylidene)chroman-4-one (12). We are aware the presence of bonds and lone electron pairs in atoms is very important, therefore we present below the bond line structures of compounds 11 and 12 according to the IUPAC principles.

   The structure of compound 11                      The structure of  compound 12

  1. Comment: Please provide more bond line structures using chemdraw or some other drawing program when discussing the synthetic scheme. The structures in Scheme 1 are very difficult to interpret. There is a pentavalent carbon in one of the structures and the authors attempt to show the 3D stereochemistry is not conventional and very difficult to understand.

Response: The authors improved the scheme 1 to make it more readable. The scheme was prepared in accordance with the ACS format. By mistake the phenyl ring was attached incorrectly, therefore the carbon atom was V-valency. We have improved the scheme as suggested by the Reviewer 4. We hope that the current version will be more friendly and understandable to the readers.

  1. Comment: Please provide more bond line structures using chemdraw or some other drawing program for each compound synthesized in the table.

Response: The authors included in the Table 3 the chemical structures for each compound.

  1. Comment: Please provide more bond line structures using chemdraw or some other drawing program next to each x-ray structure.

Response: The authors included a chemical structure of compound 2 and 8 next to the
x-ray structure of compound 2 and 8.

This manuscript is a resubmission of an earlier submission. The following is a list of the peer review reports and author responses from that submission.

Round 1

Reviewer 1 Report

The present article is devoted to the study of the cytotoxic effect of benzylidenechromanone / 3-benzylideneflavanone and their pirazolines derivatives. The search for structures with high anticancer activity and selectivity towards to cancer cells is an important task. However, the title of the article does not quite match the content.

Based on the title of the article, it is expected to see the reasons for the difference in 3-benzylidenechromanone / 3-benzylideneflavanone and their pirazolines derivatives (differences in interaction with the target, proteins, etc.). Authors should review the title or content of the article.

On page 2, the authors write (Both in this article and the previous one the compounds were tested for biological activity on four cell lines: HL-60, NALM-6, WM-115). However, there are only three lines of cancer cells.

The authors do not provide data on the selectivity of the studied compounds in relation to cancer cells. Compounds can be very toxic. Authors should make these studies.

Also, authors should compare tested compounds with anticancer drugs, for example, cisplatine, etoposide.

Since the tested compounds are previously described, section 2.1 should be moved to the Supporting Information section. Instead, the authors should in Section 3 give a detailed description of the 2D NMR spectra for each compound in order to establish their stereochemistry, taking into account the obtained XRD data. And the graphic spectra 1H, 13C and 2D NMR spectra should be presented in a readable and useful for the reader form in Supporting Information.

The authors provide a detailed description of the X-Ray data for the structures A1a and B2a. However, they do not provide an analysis of how and why structural and spatial differences affect anticancer activity. It is necessary to carry out such an analysis and draw conclusions.

In section 2.5 the authors should provide reader-friendly conclusions. Without this, the section is incomprehensible and unfinished.

On page 16 (section 3.1.), The methodology should be revised. The phrase (were heated under reflux in oil bath with mechanical stirring) is ambiguous. Maybe the authors had in mind (were heated in oil bath with mechanical stirring at the 130C for 5 hours)?

In the experimental part, the authors need to describe the procedure for the preparation of the ethereal solution of diazomethane What concentration was prepared ethereal solution of diazomethane? What molar ratio of substrate: diazomethane was used?

Authors should deeply revise the article. In its present form, it will not deserve the attention of the reader. After deep revision, the paper can be submit again.

Reviewer 2 Report

Review of Ms. Molecules-683388 entitled “Differences in cytotoxic effect between 3-benzylidenechromanone/3-benzylideneflavanone and their pirazolines derivatives” by A. A. Adamus-Grabicka et al.

In this manuscript the authors described the synthesis of ten compound and they have performed some biological studies to evaluate their potential as new anticancer drugs and to stablish some structure – activity relationships.

The cytotoxicity of the compounds under study is not very promising with the exception of  A1a. Although the results obtained for B2 and A3a are also interesting in somewhat extent. These 3 compounds  were further study in nuclease plasmid digestion and in cell cycle arrest studies. At this stage, the authors have included to compounds more A7 and B6 whose synthesis and cytotoxic potential have been described in a previous work. Their addition is not justified and no conclusions in relation to the other 10 compounds were provided. Therefore, I suggest the author to delete this part o to re-write it highlighting the interest of these new data as well as their relation with the other 10 compounds and the conclusions that could be extracted.

I do not understand the relevance of the theoretical Log P if the authors have determined it experimentally. In addition, there is a clear discrepancy between theoretical and experimental values. This section (2.3) is too long. Since the values and differences could be easily observed in Table 3, I strongly recommend the authors to summarize this part and highlight only, the SAR conclusions in terms of lipophilicity.   

The IC50 values were calculated after 48h of exposition (46h according to the experimental part) but the author have used these values to evaluate cell cycle arrest after 20h of exposure time. The authors should have evaluated cell cycle arrest after 46h of exposition or alternatively, they should have determined the IC50 values with 20h of treatment to use these concentrations. Regarding cell cycle studies, I do not appreciate differences between the vehicle control (DMSO) and 35 µM of B2. What are the p-values or the significance level of the comparisons between the different treatments? In the same way, in Fig. 8 the error bars are missing.

There is a great amount of experimental work in this manuscript but the authors should make an effort to present them. As it is, the manuscript seems to be a set of experiments put altogether without a connection between them. In addition, there is a part that it can be considered as an incremental of a published work. Therefore, I found this work unsuitable for publication in Molecules in its present form.